# Identification and Genome Characterization of a Novel Virus within the Genus *Totivirus* from Chinese Bayberry (*Myrica rubra*)

**DOI:** 10.3390/v16020283

**Published:** 2024-02-12

**Authors:** Zhongtian Xu, Yi’nan Gao, Kun Teng, Huoyang Ge, Xiaoqi Zhang, Mengjing Wu, Ruhui Li, Zujian Wu, Luping Zheng

**Affiliations:** 1Institute of Plant Virology, College of Plant Protection, Fujian Agriculture and Forestry University, Fujian 350002, China; xuzhongtian@nbu.edu.cn (Z.X.); gaoyn2023@163.com (Y.G.); tk1252658159@outlook.com (K.T.); 13838909127@163.com (H.G.); zhangxiaoqi202202@163.com (X.Z.); cyxv000@163.com (M.W.); wuzujian@fafu.edu.cn (Z.W.); 2State Key Laboratory for Managing Biotic and Chemical Threats to the Quality and Safety of Agro-Products, Institute of Plant Virology, Ningbo University, Ningbo 315211, China; 3USDA-ARS, National Germplasm Resources Laboratory, Beltsville, MD 20705, USA; ruhui.li@ars.usda.gov

**Keywords:** *Myrica rubra*, totivirus, virome

## Abstract

Chinese bayberry (*Myrica rubra*) is an economically significant fruit tree native to eastern Asia and widely planted in south-central China. However, studies about the viruses infecting *M. rubra* remain largely lacking. In the present study, we employed the metatranscriptomic method to identify viruses in *M. rubra* leaves exhibiting yellowing and irregular margin symptoms collected in Fuzhou, a city located in China’s Fujian province in the year 2022. As a consequence, a novel member of the genus *Totivirus* was identified and tentatively named “Myrica rubra associated totivirus 1” (MRaTV1). The genome sequencing of MRaTV1 was determined by overlapping reverse transcription polymerase chain reaction (RT-PCR) and rapid amplification of cDNA ends (RACE). The two deduced proteins encoded by MRaTV1 have the highest amino acid (aa) sequence identity to the coat protein (CP) and RNA-dependent RNA polymerase (RdRP) of Panax notoginseng virus A (PNVA), a member of the genus *Totivirus* within the family *Totiviridae*, at 49.7% and 61.7%, respectively. According to the results of the phylogenetic tree and the species demarcation criteria of the International Committee on Taxonomy of Viruses (ICTV) for the genus *Totivirus*, MRaTV1 is considered a new member of the genus *Totivirus*.

## 1. Introduction

Chinese bayberry (*Myrica rubra*) is an economically important fruit tree belonging to the genus *Myrica* within the family *Myricaceae*. *M. rubra*, indigenous to eastern Asia and widely planted in south-central China, is renowned for its unique flavor and medicinal value [1]. Viral diseases threaten cultivated plants, as they not only hinder their growth, leading to reduced overall yields, but also impact product quality, resulting in a decline in marketable yields [2]. Based on our field investigation, it was observed that Chinese bayberry trees were also susceptible to potential plant viral diseases during their growth cycle. However, research on the plant virus’ occurrence in Chinese bayberry was largely lacking. To date, the only research on *M. rubra* virus is the identification of a novel citlodavirus isolated from the leaves of *M. rubra* in Yunnan province, China [3]. 

Viruses within the family *Totiviridae* possess genomes consisting of a single molecule of dsRNA, ranging in length from 4.6 to 7.0 kilobase pairs (kbp) [4]. The genus *Totivirus* comprises viruses with a non-segmented, double-stranded RNA (dsRNA) genome, generally containing two open reading frames (ORFs) responsible for encoding the putative capsid protein (CP) and the RNA-dependent RNA polymerase (RdRP) [4,5]. The genus *Totivirus* contains viruses that initially infect fungi or protozoa [6,7]. However, more and more totiviruses or totivirus-like viruses have been found in a series of other eukaryotic hosts, including, but not limited to: mosquitoes [8], ants [9], fish [10], crabs [11], and plants [5,12]. The species demarcation criteria within the genus *Totivirus* is not entirely clear. In general, viruses found only in distinct host species represent different virus species, and viruses that share less than 50% of amino acid (aa) sequence identity are also considered different species (https://ictv.global/report_9th/dsRNA/Totiviridae, accessed on 6 November 2023).

Here, we conducted a metatranscriptome analysis of *M. rubra*, showing suspected viral disease symptoms which led to the identification of a novel totivirus. The genome sequence, represented by a positive genomic RNA strand of the novel totivirus, was validated by overlapping reverse transcription polymerase chain reaction (RT-PCR) and rapid amplification of cDNA ends (RACE). Phylogenetic analysis based on the RdRP sequence revealed that MRaTV1 clustered with other members of the genes *Totivirus*, suggesting that MRaTV1 represents a novel member of the genus *Totivirus*, family *Totiviridae.*

## 2. Material and Methods

### 2.1. Plant Sample Collection and RNA-Seq Sequencing Preparation

Leaves of the Chinese bayberry tree (*Myrica rubra*) showing chlorotic and notched leaf edge symptoms were collected in a park near Fujian Agriculture and Forestry University in Fuzhou (119.28° N, 26.08° W). Following collection, samples were frozen in liquid nitrogen to preserve their RNA integrity. Frozen Chinese bayberry leaf samples (approximately 5–7 leaves) were ground using a mortar and pestle in liquid nitrogen. The ground powder (about 50 mg) was subjected to total RNA extraction using the TRIzol Reagent (Invitrogen, Carlsbad, CA, USA), according to the manufacturer’s instructions. The quality and quantity of the extracted total RNAs were measured using a NanoDrop 2000C spectrophotometer (Thermo Fisher Scientific, Roskilde, Denmark). Then, the prepared RNA sample was sent to Frasergen (Wuhan, China) for transcriptome sequencing. Before sequencing, the ribosomal RNAs from the total RNAs were depleted using Ribo-Zero rRNA Removal Kit (Plant) (Illumina, San Diego, CA, USA), according to the manufacturer’s instructions. Meanwhile, RNAprep Pure Plant Kit (Polysaccharides and Polyphenolics-rich) (TianGene, Beijing, China) was also employed to obtain extracted pure RNA for sequence validation experiment. 

### 2.2. RNA Sequencing and De Novo Transcriptome Assembly

The library preparation and sequencing were conducted by Frasergen (Wuhan, China). Briefly, the library for RNA sequencing was prepared by depleting ribosomal RNAs from the total RNAs Ribo-Zero rRNA Removal Kit (Plant) (Illumina, San Diego, USA), according to the manufacturer’s instructions. Then, the indexed library was sequenced using Illumina Novaseq™ 6000 (Illumina, San Diego, CA, USA) in a paired-end manner (150 bp × 2). Raw RNA-seq datasets were processed to remove low-quality reads and adapter regions through fastp [13]. Reads with a length less than 36 nt were discarded. De novo assembly was then performed for clean reads using Trinity assembler (version 2.8.5) [14]. CD-HIT (Cluster Database at High Identity with Tolerance) was used with the parameter (−c 0.95) to cluster similar biological sequences together, reducing sequence redundancy [15].

### 2.3. Viral Contigs Identification, Calculations

To identify and annotate the virus-associated contigs, the assembled transcriptome contigs were blasted against the NCBI (https://www.ncbi.nlm.nih.gov, accessed on 3 May 2023) NR database using the BLASTx program [16]. After annotation, clean reads were aligned to the candidate viral contigs using the Burrows–Wheeler Aligner (BWA) program [17] with default parameters.

### 2.4. Overlapping RT-PCR and RACE

To verify the assembled viral sequence, specific primers (Appendix A) were designed to get the amplification products of the target virus with the amplicon size ranging from about 752 bp to 1074 bp. The primer specificity was checked by Primer-BLAST (https://www.ncbi.nlm.nih.gov/tools/primer-blast). The NCBI’s Primer-BLAST online service was used to identify the optimal primer pairs, and then primer pairs were synthesized by BGI Tech (Beijing, China). RT-PCR was conducted using FastKing One-Step RT-PCR Kit (TianGene). The 5′-terminus was determined using a 5′ RACE System for Rapid Amplification of cDNA Ends (Version 2.0) (Invitrogen, Waltham, MA, USA). RT-PCR reactions were performed in a 25-μL reaction containing 1.0 μL total RNA, 1.0 μL forward primer (10 μM ), 1.0 μL reverse primer (10 μM), 12.5 μL 2× FastKing One Step RT-PCR MasterMix, 1μl of 25×RT-PCR Enzyme Mix, and 8.5 μl of water. The thermal cycling conditions for RT-PCR were as follows: 42 °C for 30 min, 95 °C for 3 min, then 94 °C for 30 s, 53–60 °C for 30 s, 72 °C for 1 min (35 cycles), and 72 °C for 5 min. After PCR amplification, the products were separated via gel electrophoresis on a 1.5% agarose gel in the presence of a TBE buffer. The PCR products were stained using ethidium bromide, and visualized under ultraviolet light. Finally, PCR products were purified and cloned into pGM-T vector (TianGene), and plasmid DNAs isolated from selected colonies (at least 6 for each amplicon) were sequenced by Tsingke (Beijing, China).

### 2.5. Viral Sequence Analysis

The ORF Finder online server (https://www.ncbi.nlm.nih.gov/orffinder, accessed on 6 May 2023) was used to predict MRaTV1’s open reading frame (ORF). The conserved domains of MRaTV1 were predicted by both NCBI CD-search (https://www.ncbi.nlm.nih.gov/Structure/cdd/wrpsb.cgi, accessed on 6 May 2023) and InterProScan (https://www.ebi.ac.uk/interpro/, accessed on 6 May 2023). To obtain the transcriptome coverage alone MRaTV1 genome, the clean reads were mapped to the MRaTV1 genome using BWA, and the sequencing depth on each site was used to generate the viruses’ transcript coverage plots.

### 2.6. Construction of Phylogenetic Trees

For analysis of the newly identified MRaTV1, the amino acid (aa) sequences of MRaTV1 RdRP, together with other representative members of genus *Totivirus*, *Victorivirus,* and *Leishmaniavirus,* the *Giardiavirus* of family *Totiviridae* retrieved from the NCBI nucleotide database were aligned using MAFFT (version 7.0) [18]. Next, trimAl was used for the removal of spurious sequences or poorly aligned regions from the multiple sequence alignments [19]. Then, IQ-TREE (v1.6.6) [20] was utilized for conducting a phylogenetic analysis based on the maximum likelihood method with the best-fit amino substitution model selected by ModelFinder [21]. To assess the confidence in the topology, 5000 Ultrafast bootstrap replicates were employed.

## 3. Results and Discussion

To investigate potentially causing pathogens, the leaves of the Chinese bayberry tree (*Myrica rubra*) exhibiting chlorotic and notched leaf edge symptoms were collected and recorded (Figure 1A,B). Leaves showing disease symptoms were ground in liquid nitrogen and used to extract the total RNAs for metatranscriptome sequencing. Before transcriptome sequencing, ribosome-RNAs were depleted from the sample. The rRNA-depleted total RNA sequencing produced a total of 99,450,902 raw reads. Following quality control procedures, 98,298,500 clean reads were retained and subjected to the de novo assembly. After the de novo assembly process and removal of the redundancy contigs, 200,274 contigs were retained for the downstream analysis (max length: 16,236 bp, min length: 200 bp, N50: 635 bp). The assembled sequences obtained from the de novo assembly were subjected to the BLASTx search against the NCBI NR database to identify the potential viral contigs. As a result, two protein sequences deduced from the assembled sequence are found to be annotated with viruses within the family *Totiviridae* with a credible similarity. Interestingly, when the corresponding viral contigs were subjected to a BLASTn search in the NCBI NT database, no significant matches or similarities were found at the nucleotide level, suggesting that these contigs may represent the novel viral sequence. 

To confirm the presence of RNAs corresponding to the assembled contigs, potentially representing novel viruses, we conducted RT-PCR assay. The RT-PCR results evidenced these RNAs, and subsequent Sanger sequencing validated the correctness of the assembly results. Then, RACE was also implemented to get the termini of the viral RNA. After obtaining the potential novel viral RNA’s maximum length possible, overlapping RT-PCR assays were carried out for the original *M. rubra* samples using virus full-length-specific primers (Appendix A) to display the result. As shown in Figure 1C, a series of specific primers were used, each designed to target different regions of the potential viral genome. These primer sets were expected to produce amplicons with lengths ranging from 752 bp to 1074 bp. The gel electrophoresis image depicted in Figure 1C displays that all these specific primers produced specific bands with corresponding lengths. Sanger sequencing results of the PCR product are identical to the corresponding assembled sequence region, which verifies the accuracy of metatranscriptome sequencing. It should be noted that we also made attempts to confirm the terminal sequence of MRaTV1 genome using both 5′RACE and 3’RACE methods. However, only 5′RACE was successful, while, despite multiple attempts, we could still not obtain the 3′RACE product. Nevertheless, our confirmed 3′ untranslated region (3’UTR) of MRaTV1 genome is of reasonable length compared to other totiviruses (Table 1). The related RT-PCR and RACE primers are listed in Appendix A. 

The nearly complete MRaTV1 genome spans a length of 4704 nt. It consists of two non-overlapping open reading frames (ORFs) on the positive strand which are separated by a 145 bp intergenic region, as revealed by ORFfinder. Untranslated regions (UTRs) of 84 nt and 151 nt were detected at the 5′ and 3′ end, respectively. For comparison, its best hit in BLASTx search, Panax notogiseng virus A (PnVA) (accession: NC_029096.1), has a genome length of 5003 nt and 5′UTR, 3′UTR, and an intergenic region of PnVA measuring 77 nt, 64 nt, and 114 nt, respectively (Table 1). The newly identified MRaTV1 was found to share a similar genome organization to PnVA, i.e., encoding a CP protein and an RdRP protein in sequential order from 5′ termini to 3′ termini. The length of the major coat protein domain of MRaTV1 (386 aa) is approaching that of PnVA (387 aa). Interestingly, although the ORF2 of MRaTV1 (2175 nt) is shorter than that of PnVA (2469 nt), the length of the RdRP domain of MRaTV1 (446 aa) is larger than that of PnVA (442 aa). The blastp analysis revealed a similarity of 49.7% between amino acid sequence of MRaTV1 ORF1 and PnVA ORF1, whereas MRaTV1 ORF2 exhibited a higher similarity of 61.7% with PnVA ORF2. When we aligned the clean reads back to the MRaTV1 genome, 5080 reads could be mapped back to the MRaTV1 genome. The genome of MRaTV1 had nearly full coverage by viral reads (Figure 1D).

To ascertain the evolutionary position of MRaTV1, a phylogenetic analysis was conducted using the RdRp proteins of MRaTV1 and representative members from four genera of the family *Totiviridae*. These genera include the genus *Totivirus*, *Victorivirus*, *Leishmaniavirus*, and *Giardiavirus*. The results reveal that members of distinct genera form distinctive subclades, and MRaTV1 is positioned on a sub-branch within the genus *Totivirus* in the evolutionary tree. The closest phylogenetic taxon to MRaTV1 is PnVA, and the ultrafast bootstrap support value labeled on the node indicates high reliability for the evolutionary tree. Interestingly, we observed that within the evolutionary branch of the genus *Totivirus*, scheffersomyces segobiensis virus L (accession number: NC_038697.1) and saccharomyces cerevisiae virus La (accession number: U01060.1) form a distinct sub-branch. Both of these two viruses infect fungal hosts. Another sub-branch consists of puccinia striiformis totivirus 1 (accession number: KY207361.1), puccinia striiformis totivirus 2 (accession number: KY207362.1), and xanthophyllomyces dendrorhous virus L2 (accession number: JN997474.2), all of which infect fungal hosts as well. Besides that, In the phylogenetic tree, panax notoginseng virus A (accession number: KT388111.1), black raspberry virus F (accession number: NC_009890.1), ribes virus F (accession number: EU495331.1), and MRaTV1 cluster together as a monophyletic group (Figure 2). This result indicates that MRaTV1 should be a virus associated with the *M. rubra* plant rather than a symbiotic microorganism such as fungi in *M. rubra*.

As discussed in a paper published in the year 2023, the presence of highly diverse microbial communities comprising fungi, bacteria, archaea, and protists colonizing plants and forming symbiotic relationships emphasizes the need for careful examination of the origin of viruses detected in viromic studies [22]. The genus *Totivirus* contains viruses that initially infect fungi or protozoa [5,6,7]. For a long time, viruses within the genus *Totivirus* were believed to only infect fungi or protozoa. However, numerous viromic studies have revealed an increasingly diverse range of hosts for viruses belonging to the genus *Totivirus*. Interestingly, multiple mycoviruses have been identified from the symbiont *Pestalotiopsis* spp. isolated from *M. rubra* [23]. This raises our concern regarding whether the identified host of the novel totivirus in this study is *M. rubra* itself or potentially fungi parasitizing *M. rubra*. However, no symptoms related to fungal infection were observed in the *M. rubra* leaves we collected. Additionally, the brightness of the RT-PCR band contradicts the hypothesis that the novel totivirus’s host is an endophytic fungus in *M. rubra* (Figure 1C). Besides, Panax notoginseng virus A (accession number: KT388111.1), Ribes virus F (accession number: EU495331.1), and MRaTV1 form a monophyletic branch in the phylogenetic tree (Figure 2). All of the above results support the conclusion that the newly identified totivirus’s host is *M. rubra* itself, rather than the symbiont fungi. MRaTV1 is the first totivirus identified in *M. rubra*. According to the species demarcation criteria within the genus *Totivirus*, we propose that MRaTV1 is the new member of the genus *Totivirus*.

## 4. Conclusions

While *Myric rubra* has long been cultivated as a significant economic tree species, research on virus diseases affecting *M. rubra* has been notably scarce. This study introduces a novel totivirus, designated MRaTV1, identified in *M. rubra*. This is the first report of a totivirus in *M. rubra*, and its experimentally confirmed genome has been deposited in the NCBI nucleotide database under the accession number OR915853. By characterizing this new totivirus, our research not only enriches the catalog of documented viruses associated with *M. rubra*, but also expands the known host range of totiviruses. The experimental validation of the genome sequence facilitates a deeper understanding of the genomic structure and phylogenetic position of this newly discovered totivirus.

## Figures and Tables

**Figure 1 viruses-16-00283-f001:**
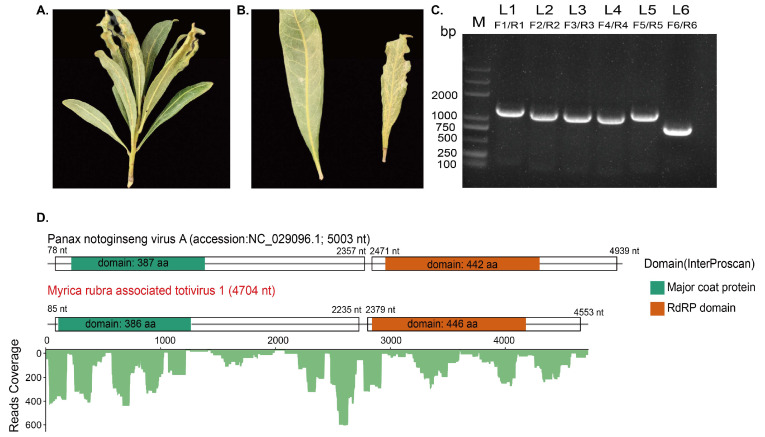
Identification of genome characterization of *Myrica-rubra*-associated totivirus 1 (MRaTV1) in Chinese bayberry. (**A**) Twig showing yellowing and notched leaf edge symptoms. (**B**) Leaves showing disease symptoms. (**C**) Overlapping RT-PCR of newly identified MRaV1. M: marker, Genstar marker DL2000. L1–L6: overlapping amplicons. (**D**) Genome structure and transcriptome clean read coverage along the genome of MRaTV1. The reference genome represents the best hit in the BLASTx search on the upper board (with black font), and the novel virus identified in Chinese bayberry is on the lower panel (marked red). The grey line in the diagram represents the virus genome. Each box represents an open reading frame (ORF) of the virus. Conserved functional domains are color-coded, and the corresponding names are shown on the figures’ right side.

**Figure 2 viruses-16-00283-f002:**
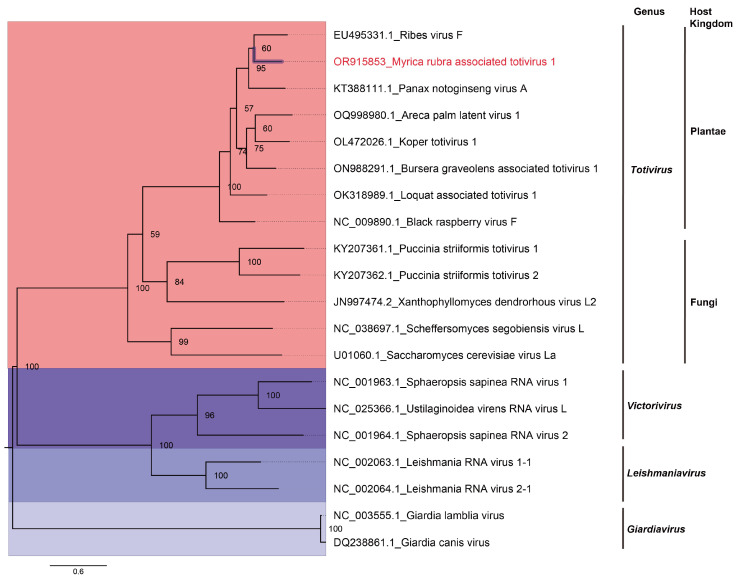
Phylogenetic analysis of MRaTV1 identified in Chinese bayberry. The maximum-likelihood phylogenetic midpoint-rooted tree was inferred based on the RdRP protein sequence of representative members of different genera in Tovirividae retrieved from NCBI and MRaTV1 identified in this research. The bar represents the number of substitutions per site (0.6). Myrical rubra associated totivirus 1 (MRaTV1) identified in this study is in red font. The four colored boxes from top to bottom represent the genus *Totivirus, Victorivirus, Leishmaniavirus,* and *Giardiavirus*, respectively. Ultrafast bootstrap support value (5000 replicates) was calculated and given at each node. The model selected by ModelFinder is LG+F+I+G4.

**Table 1 viruses-16-00283-t001:** Comparison of length of CP, RdRP, 5′UTR, 3′UTR, and IR among different totiviruses.

Virus	Genome (nt)	CP (aa)	RdRP (aa)	5′UTR (nt)	3′UTR (nt)	IR ^1^ (nt)	GC (%)
MRaTV1	4704	716	724	84	151	126	51.15
^2^ PnVA	5003	760	823	77	64	114	43.11
^3^ BgTV1	4794	687	767	67	57	309	43.14
^4^ BrVF	5077	770	815	198	78	147	45.48

^1^ IR: intergenic region. ^2^ PnVA: Panax notogiseng virus A (accession: NC_029096.1). ^3^ BgTV1: Bursera-graveolens-associated totivirus 1 (accession: ON988291). ^4^ BrVF: black raspberry virus F (accession: NC_009890.1).

## Data Availability

The genome sequence of MRaTV1 was deposited in the NCBI GenBank database under accession numbers OR915853.

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
