# Peer review of "Identification and Genome Characterization of a Novel Virus within the Genus Totivirus from Chinese Bayberry (Myrica rubra)"

_viruses, 2024, doi:10.3390/v16020283_

Round 1

Reviewer 1 Report

Comments and Suggestions for Authors

The brief report entitled "Identification and genome characterization of a novel virus within the genus Totivirus from Chinese bayberry (Morrella rubra) by Xu and colleagues reports on the collection of a leaf (or several leaves?) from a symptomatic tree, RNA extraction and identification of a new virus belonging to the genus Totivirus. They identified the two typical ORFs of this type of viruses, the capsid encoding ORF (sometimes referred to as Gag) and the polymerase encoding sequence. This new virus increases the number of totiviruses infecting non-fungal organisms. It also differs from typical totiviruses by the presence of an intergenic region (totiviruses mainly present overlapping ORFs).

Although this report is worth publishing, I have a few concerns and questions that should first be considered.

The authors consider they have conducted a metagenomic analysis however I am not sure that a few leaves on a tree can really be considered an ecological environment. Especially since they searched NCBI NR database only to identify potential viral contigs. Thus, it rather seems that they tried to establish the virome of a few symptomatic leaves. Please correct this overstatement.

Lines 53-55: the authors claim they have validated the genome sequence by RT-PCR and RACE. I do not understand how a dsRNA genome can be validated by techniques applicable also to ssRNA. Most of the primer pairs allow to confirm the presence of mRNAs corresponding to their assembled contigs. It should be better explained what the authors mean through "validate the genome sequence". The detection of dsRNA or at least of the minus strand would be a much better validation. Actually from the results it is not clear what RNAs were really detected. This informtion could be added in Fig. 1.

Line 61: Was only one leaf collected? This seems the case also from line 128, however from lines 63 and 130, one rather understands several samples were collected. How many leaves from how many trees were collected? What does a sample represent: a single leaf? If so, please specify how many samples were collected.

§ 2.1 & 2.2. The protocol for total RNA extraction is not clear. Why are two methods described, Trizol and Pure plant Kit? Were they both used on the same samples? Why? It is also not quite clear whether the extraction separates or not ssRNAs from dsRNAs? What do the authors mean by transcriptome sequencing? Are only transcripts sequenced (after polyA selection)? I guess there is no way to discriminate between viral transcripts and viral genomes.

Line 52: a virus can not be amplified in a tube, only cells can amplify viruses.

Line 96: What on was the RT-PCR conducted? the same total RNA extracts that were used for the sequencing?

Lines 110-115: What was analyzed using the ORF finder online server? The sequences from the cloned amplicons or the RNA-Seq reads? It is not quite clear what sequence is considered the genome of MRaTV1 the sequence of assembled contigs or the sequence of the amplicons? How much do these  sequences differ?

Line 155: contigs are not present in a biological sample, they are assembled sequences on a computer. Thus, it is not clear what is looked for in the samples: transcripts allowing protein translation or the dsRNA genome of the virus? I guess only primer pairs overlapping the intergenic region or the two ORFs allow for the detection of the viral genome whereas other pairs do not discriminate between transcripts and genomic RNAs? This should be better explained throughout the manuscript.

Lines 164-168: 5'RACE and 3'RACE are generally intended to identify mRNA ends. What RNA was used for the RACE experiments? Were the RACE experiments conducted to identify the genome ends? This should be clarified as the MRaTV1 itself (virus) has no terminal sequence, only its genome or transcripts have terminal sequences.

Lines 183-184: Formally, similarity is fine for aminoacids whereas ORFs refer to nt sequences, please correct.

Lines 214-232: what is known about genomic sequences of endophytic fungi of M. rubra? if such a fungus was present in the plant, could it be seen from reads matching known sequences?  Or were only viral contigs looked for? Were many reads obtained in this study unassigned, I mean not corresponding to M. rubra neither to viruses?

Conclusion: There is some redundancy.

Fig 1d: a space is needed between a number and its unit, please add a space before aa and before nt. The position of the primers used to confirm the genomic organization should be indicated on the figure, this would help understanding the validation of the genome (vs transcript detection)

Supp table 1:

It is not clear what footnote 2 refers to. Have not all primers of the table been used for RT-PCR amplification as stated in the title?

It is not clear what fragment of Fig 1c L1 to L6 correspond to what fragment of given length in the table, an additional column with the name of the amplicon used in Fig 1c could be added to the table.

Also, Fig 1c does not show any amplicon of 274 bp; better explain the primer pairs used to obtain the amplicons of 420 and 274 bp.

If primer -R1 was used with primer -F1, the amplicon size of 1,74 bp is fine however -F2 with -R2 should give a 986 bp fragment, -F3 with -R3 is expected to give a 1,004 bp fragment, -F4 with -R4 a 902 bp amplicon, -F5 with R-5 a 998 bp fragment and -F6 with -R6 a 754 bp fragment. Please check primer positions and amplicon size or better explain what is expected.

Reference list: remove extra capital letters from titles.

Comments on the Quality of English Language

A few typography or agreement errors need to be corrected

Author Response

Responses to Reviewer #1

Reviewer 1: General Comments:

The brief report entitled "Identification and genome characterization of a novel virus within the genus Totivirus from Chinese bayberry (Morrella rubra) by Xu and colleagues reports on the collection of a leaf (or several leaves?) from a symptomatic tree, RNA extraction and identification of a new virus belonging to the genus Totivirus. They identified the two typical ORFs of this type of viruses, the capsid encoding ORF (sometimes referred to as Gag) and the polymerase encoding sequence. This new virus increases the number of totiviruses infecting non-fungal organisms. It also differs from typical totiviruses by the presence of an intergenic region (totiviruses mainly present overlapping ORFs).

Although this report is worth publishing, I have a few concerns and questions that should first be considered.

The authors consider they have conducted a metagenomic analysis however I am not sure that a few leaves on a tree can really be considered an ecological environment. Especially since they searched NCBI NR database only to identify potential viral contigs. Thus, it rather seems that they tried to establish the virome of a few symptomatic leaves. Please correct this overstatement.

Response: We greatly appreciate your positive comments and suggestions for revising our paper! The points you raised show that you have a deep understanding of the foundational knowledge related to totivirus and have thoroughly inspected our manuscript. We apologize for some typos and lack of clarity in our manuscript. We did collect some symptomatic Chinese bayberry twigs, and metatranscriptomic sequencing was performed on several leaves. As you have pointed out this issue in the minor comments part, we will provide a comprehensive response and detailed feedback in that section. In your review comments, you questioned the use of “metagenomics”. However, the term “virus metagenomics” is accepted in the scientific community, as evidenced by its usage in the following links 1, 2, 3, and 4. Nevertheless, we acknowledge your perspective. If the primary focus of the research is solely on viruses, the term “metagenomics” might indeed appear somewhat overstated. We appreciate your valuable comments once again, as they significantly contribute to improving the overall quality of our manuscript. I will respond to your comments in a point-by-point way in the following. 

1, DOI: 10.1094/PHYTO-12-14-0356-RVW

2, DOI: 10.1146/annurev-genet-110711-155600

3, DOI: 10.3390/life12122048

4, https://www.mdpi.com/journal/viruses/special_issues/mev2

Reviewer 1: Minor Comments:

- Lines 53-55: the authors claim they have validated the genome sequence by RT-PCR and RACE. I do not understand how a dsRNA genome can be validated by techniques applicable also to ssRNA. Most of the primer pairs allow to confirm the presence of mRNAs corresponding to their assembled contigs. It should be better explained what the authors mean through "validate the genome sequence". The detection of dsRNA or at least of the minus strand would be a much better validation. Actually from the results it is not clear what RNAs were really detected. This informtion could be added in Fig. 1.

Response: For dsRNA viruses, only a single strand is typically archived in the NCBI nucleotide database to represent their genome sequence. Nevertheless, we should not interpret the presence of only a single sequence as indicative of an incomplete genome sequence. But which strand represents the genome sequence should be clarified, we have added the detail in the revised manuscript. Please see L56. 

- Line 61: Was only one leaf collected? This seems the case also from line 128, however from lines 63 and 130, one rather understands several samples were collected. How many leaves from how many trees were collected? What does a sample represent: a single leaf? If so, please specify how many samples were collected.

Response: We apologize for our unclear statement. In fact, we collected several twigs with leaves showing disease symptoms from the same Chinese bayberry tree. After the sample collection, approximately 5~7 leaves were selected for RNA extraction and high-throughput sequencing. We have added the relevant details in the revised manuscript. Please see L63 and L66-L67.

- § 2.1 & 2.2. The protocol for total RNA extraction is not clear. Why are two methods described, Trizol and Pure plant Kit? Were they both used on the same samples? Why? It is also not quite clear whether the extraction separates or not ssRNAs from dsRNAs? What do the authors mean by transcriptome sequencing? Are only transcripts sequenced (after polyA selection)? I guess there is no way to discriminate between viral transcripts and viral genomes.

Response: The Chinese bayberry leaves were collected in early 2022. After the high-throughput sequencing, we have made numerous attempts to obtain the full genome sequence of this novel totivirus. In our experiments, we did use both two RNA extraction methods simultaneously. The Trizol method can generate more RNA for high-throughput sequencing (recommended by NGS sequencing company). Meanwhile, the Pure Plant Kit was employed to produce purer RNA for sequence validation experiments. We did not separate ssRNAs from dsRNAs before the metatranscriptome sequencing. In fact, we don’t fully understand how this “separation” could be implemented. We used the rRNA depletion method instead of polyA selection, which ensures that more viral reads could be obtained. In terms of sequencing methods, there is no way to discriminate between viral transcripts and viral genomes indeed. However, the presence of viral transcripts-derived reads and viral genome-derived reads could contribute to the recovery of the virus genome sequence in bioinformatics analysis. We have rephrased the paragraph for clarity. Please see L72-L77 in the revised manuscript.

- Line 52: a virus can not be amplified in a tube, only cells can amplify viruses.

Response: Thank you for your insightful comment. We appreciate your attention to detail. To clarify, we acknowledge that viruses cannot replicate in a tube. Our statement regarding the amplification of the viral sequence refers to the PCR amplification of the specific genomic regions using the designed primers. We have rephrased the sentence for clarity. Please see L97-L98 in the revised manuscript.

- Line 96: What on was the RT-PCR conducted? the same total RNA extracts that were used for the sequencing?

Response: In our actual experiments, after the extraction of RNA, half of RNA was sent to NGS sequencing company for metatranscriptome sequencing, and half was kept for sequence validation experiment. So we can reply “yes” to your question. However, as I have mentioned above, we have conducted numerous attempts to get the full length of this novel totivirus (mainly stuck at the obtaining of 3’ termini sequence). After the run out of the preserved RNA, we also extracted the RNA from the remaining samples for sequence validation experiment.

- Lines 110-115: What was analyzed using the ORF finder online server? The sequences from the cloned amplicons or the RNA-Seq reads? It is not quite clear what sequence is considered the genome of MRaTV1 the sequence of assembled contigs or the sequence of the amplicons? How much do these sequences differ?

Response: The near-full genome length of MRaTV1, obtained by the assembly of overlapping RT-PCR amplicons and RACE amplicon, was subjected to ORF finder for open reading frame prediction. In fact, we verified the genome sequence we finally obtained through six overlapping RT-PCR amplification amplicons, as shown in Figure 1C. In our understanding, the assembly result of those amplicons represents the full length of MRaTV1, while each amplicon represents only a part of the genome fragment.

- Line 155: contigs are not present in a biological sample, they are assembled sequences on a computer. Thus, it is not clear what is looked for in the samples: transcripts allowing protein translation or the dsRNA genome of the virus? I guess only primer pairs overlapping the intergenic region or the two ORFs allow for the detection of the viral genome whereas other pairs do not discriminate between transcripts and genomic RNAs? This should be better explained throughout the manuscript.

Response: We appreciate your attention to detail. We appreciate the reviewer's comments and acknowledge that our statement could have been clearer. It is important to note that the RT-PCR assay as shown in Figure 1C depicts the amplification results based on the determined full length of the MRaTV1 genome. In your review comments, you have been trying to distinguish between viral transcript RNAs and viral genomic RNAs. However, we feel that this distinction is not that important. In terms of obtaining the final viral genomic sequence, which type of RNA is served as template will not affect the correctness of the sequence. We have rephrased the paragraph to make the statement clear, please see L161-L167.

- Lines 164-168: 5'RACE and 3'RACE are generally intended to identify mRNA ends. What RNA was used for the RACE experiments? Were the RACE experiments conducted to identify the genome ends? This should be clarified as the MRaTV1 itself (virus) has no terminal sequence, only its genome or transcripts have terminal sequences.

Response: In addition to mRNA, the 5’ and 3’ ends of the genomic RNA of RNA viruses can also be determined through RACE. The RACE experiments were conducted to get the genome ends of the novel totivirus indeed. In response to this comment “This should be clarified as the MRaTV1 itself (virus) has no terminal sequence, only its genome or transcripts have terminal sequences.”, we have made the corresponding modifications in L174 and L177.

- Lines 183-184: Formally, similarity is fine for aminoacids whereas ORFs refer to nt sequences, please correct.

Response: Thank you for bringing this to our attention. We have corrected the sentence in L198 on revised manuscript.

- Lines 214-232: what is known about genomic sequences of endophytic fungi of M. rubra? if such a fungus was present in the plant, could it be seen from reads matching known sequences?  Or were only viral contigs looked for? Were many reads obtained in this study unassigned, I mean not corresponding to M. rubra neither to viruses?

Response: To be honest, during the 10-day revision period allotted to us, the majority of the time was spent figuring out how to respond to this question. Currently, there is very limited research on the endophytic fungi in M. rubra, not to mention the availability of an endophytic fungi’s reference genome for analysis. Furthermore, due to the absence of a well-assembled genome for Myrica rubra at present, it is also difficult to determine the proportion of reads associated with M. rubra. The number of reads corresponding to MRaTV1 is 5,080. We attempt to align our assembly results with some known fungal conserved Internal Transcribed Spacer (ITS) RNA sequences. However, no blast hits were obtained in our attempts. Combining with other evidence presented in our paper, we believe that there is no presence of so called fungi or endophytic fungi in the samples.

- Conclusion: There is some redundancy.

Response: Done. We have adjusted the paragraph to improve it. Please see L251 to L259.

- Fig 1d: a space is needed between a number and its unit, please add a space before aa and before nt. The position of the primers used to confirm the genomic organization should be indicated on the figure, this would help understanding the validation of the genome (vs transcript detection)

Response: Thank you very much for noticing such details. We have made corresponding revisions based on your review comments. Please see the Figure 1 in the revised manuscript.

- Supp table 1:

It is not clear what footnote 2 refers to. Have not all primers of the table been used for RT-PCR amplification as stated in the title?

It is not clear what fragment of Fig 1c L1 to L6 correspond to what fragment of given length in the table, an additional column with the name of the amplicon used in Fig 1c could be added to the table.

Also, Fig 1c does not show any amplicon of 274 bp; better explain the primer pairs used to obtain the amplicons of 420 and 274 bp.

If primer -R1 was used with primer -F1, the amplicon size of 1,74 bp is fine however -F2 with -R2 should give a 986 bp fragment, -F3 with -R3 is expected to give a 1,004 bp fragment, -F4 with -R4 a 902 bp amplicon, -F5 with R-5 a 998 bp fragment and -F6 with -R6 a 754 bp fragment. Please check primer positions and amplicon size or better explain what is expected.

Response: We sincerely apologize for this kind of basic mistake. Once again, we express our sincere gratitude to you for your thorough review of our manuscript! Actually, footnote 2 refers to nothing in the table. We have removed the extra footnote 2. We added the corresponding amplicon tags for each primer pair. Please take a look at the revised Supplementary table. The problems with the positions in Supplementary Table 1 primarily stem from a one-base displacement. We have double-checked the position of each primer pair. Please see the revised Supplementary Table 1.

- Reference list: remove extra capital letters from titles.

Response: Done. Please check the reference list in the revised manuscript.

Reviewer 2 Report

Comments and Suggestions for Authors

Authors Zhongtian Xu and co-workers presented here a manuscript of a brief report entitled “Identification and genome characterization of a novel virus within the genus Totivirus from Chinese bayberry (Morrella rubra)”.

The main contribution of the paper is the description of a novel totivirus from the symptomatic plant Myrtilla rubra, identified using HTS applied to these plants.

The authors obtained the complete genomic sequence of a new virus and performed a phylogenetic analysis to show its relationship to the genus Totivirus.

Their findings are novel as they describe a new virus.

The paper is well written and the methods and procedures used are well described.

The results are presented in sufficient detail. The phylogenetic analysis is well done.

The discussion is consistent and addresses potential problems with considering totiviruses as viruses that only infect fungi.

I have one serious comment: first of all, the plant mentioned as a virus host has corect name Myrica rubra. Please unify the name of the plant and the virus throughout the manuscript.

Minor comments:

lines 55-58 belong to results

line 58: Totivirius – should be “Totivirus”

line 64: why pumpkin?

line 75: missing word: from the total RNAs using Ribo-Zero rRNA Removal

lines 119-120: virus genera should be in italics

line 128: leaf – should be leaves

line 128-132 belong more to Material and Methods; description of observed symptoms should remain here

Table 1: virus acronyms must be explained

lines 184-185: please give the exact number of reads covering the genome

line 199: In the phylogenetic tree – should not be capitalised

As the authors have provided two supplementary files, these should be listed in an appropriate paragraph at the end of the manuscript.

The paper, after minor revision, is worthy of publication in the journal Viruses.

Comments on the Quality of English Language

Minor problems listed above.

Author Response

Responses to Reviewer #2

Reviewer 2: General Comments:

Authors Zhongtian Xu and co-workers presented here a manuscript of a brief report entitled “Identification and genome characterization of a novel virus within the genus Totivirus from Chinese bayberry (Morrella rubra)”.

The main contribution of the paper is the description of a novel totivirus from the symptomatic plant Myrtilla rubra, identified using HTS applied to these plants.

The authors obtained the complete genomic sequence of a new virus and performed a phylogenetic analysis to show its relationship to the genus Totivirus.

Their findings are novel as they describe a new virus.

The paper is well written and the methods and procedures used are well described.

The results are presented in sufficient detail. The phylogenetic analysis is well done.

The discussion is consistent and addresses potential problems with considering totiviruses as viruses that only infect fungi.

I have one serious comment: first of all, the plant mentioned as a virus host has corect name Myrica rubra. Please unify the name of the plant and the virus throughout the manuscript.

Response: We are very grateful for your positive comments of our work! Currently, there is inconsistency in the correct Latin name for the Chinese bayberry. Some sources state it as “Morella rubra,” while others suggest “Myrica rubra.” As you have suggested, “Myrica rubra” is used more often in academic papers, as evidenced in the research articles (DOI: 10.1186/s12864-019-5818-7; DOI: 10.1007/s00705-023-05762-1). We have unified the name of the plant and virus in the revised manuscript.

Reviewer 2: Minor Comments:

- lines 55-58 belong to results.

Response: While reviewing our manuscript, we greatly agree with your comment that this section indeed belongs to the results part. However, we kindly request permission to retain this content. For a brief report, we aim to quickly present the main results in the introduction of the whole manuscript.  

- line 58: Totivirius – should be “Totivirus”

Response: We express our sincere gratitude to you for your thorough review of our manuscript! We have corrected the typo in the revised manuscript. Please check the L60.

- line 64: why pumpkin?

Response: We have made corrections for this issue in the revised manuscript. It should be Chinese bayberry. Please check the L66-L67. 

- line 75: missing word: from the total RNAs using Ribo-Zero rRNA Removal

Response: We have added the missing information and made corrections for this issue in the revised manuscript. Please check the L72-L75.

- lines 119-120: virus genera should be in italics

Response: Done. Please check the L125 in the revised manuscript.

- line 128: leaf – should be leaves

Response: Done. Please check the L134 in the revised manuscript.

- line 128-132 belong more to Material and Methods; description of observed symptoms should remain here

Response: We agree with your comment, but we also maintain a certain reservation. Description of observed symptoms of the collected samples is, to some extent, a form of a result as well. There is another issue: if we discuss sample symptoms in the Material and Methods part, we may need to change the position of Figure 1 in the manuscript. This would lead to Figure 1 appearing too early in the manuscript. If we separate Figure 1 into two independent figures, the final number of figures would exceed the maximum allowable number of figures for a brief report (Only 2 figures are allowed).

- Table 1: virus acronyms must be explained

Response: Done, please check the footnotes in Table 1.

- lines 184-185: please give the exact number of reads covering the genome

Response: We added the missing information in the revised manuscript. Please see the L200-L201.

- line 199: In the phylogenetic tree – should not be capitalized

Response: We are not entirely sure if we fully understand this comment, but we have removed the capitalization of the first letter of the full names of all viruses in that paragraph, as we have done in the discussion about phylogenetic tree in our another published paper (DOI: 10.1007/s00705-023-05944-x). Please check the revise manuscript in L211 to L218.

- As the authors have provided two supplementary files, these should be listed in an appropriate paragraph at the end of the manuscript.

Response: We have listed the two supplementary files at the end of the manuscript. Please take a look at L261.

Round 2

Reviewer 1 Report

Comments and Suggestions for Authors

The authors still confuse the virus and its RNA.

Line 161: formally, contigs are not present elsewhere as in the computer thus, please replace with "to confirm the presence of RNAs corresponding to the assembled contigs"

Line 162: change to "The RT-PCR results evidenced these RNAs"

Line 164: the virus is a nucleocapsid and thus has no termini, please change to "to get the termini of the viral RNA"

Line 165: the virus is isometric, it has no maximum length. Only its genome/RNA has the maximum length mentioned by the authors

Author Response

Responses to Reviewer #2 in Round 2

General response: We would like to thank you for your careful review of our manuscript once again! Upon careful reading of your detailed and valuable comments, we have indeed realized the inaccuracy in our expression in the manuscript. In our following work, we will pay more attention to the distinction between the expression of "virus" and "its RNAs".

- Line 161: formally, contigs are not present elsewhere as in the computer thus, please replace with "to confirm the presence of RNAs corresponding to the assembled contigs"

Response: Done! Please see L161.

- Line 162: change to "The RT-PCR results evidenced these RNAs"

Response: Done! Please see the L162 to L163.

- Line 164: the virus is a nucleocapsid and thus has no termini, please change to "to get the termini of the viral RNA"

Response: Done! Please take a look at L164.

- Line 165: the virus is isometric, it has no maximum length. Only its genome/RNA has the maximum length mentioned by the authors

Response: Thank you for your insightful comment! We have corrected our expression in our manuscript. Please take a look at L165.